# Integrated Steganography and Steganalysis with Generative Adversarial Networks

## Abstract

Recently, generative adversarial network is the hotspot in research areas and industrial application areas. Its application on data generation in computer vision is most common usage. This paper extends its application to data hiding and security area. In this paper, we propose the novel framework to integrate steganography and steganalysis processes. The proposed framework applies generative adversarial networks as the core structure. The discriminative model simulate the steganalysis process, which can help us understand the sensitivity of cover images to semantic changes. The steganography generative model is to generate stego image which is aligned with the original cover image, and attempts to confuse steganalysis discriminative model. The introduction of cycle discriminative model and inconsistent loss can help to enhance the quality and security of generated stego image in the iterative training process. Training dataset is mixed with intact images as well as intentional attacked images. The mix training process can further improve the robustness and security of new framework. Through the qualitative, quantitative experiments and analysis, this novel framework shows compelling performance and advantages over the current state-of-the-art methods in steganography and steganalysis benchmarks.

## 1    Introduction

Steganography literally means "covered writing" and is usually interpreted to hide information in other information. As the counterpart, the main idea of steganalysis is to analyze whether the received information contains any hidden information, and to recover the hidden information if possible (Volkhonskiy et al., 2017). Since their birth, steganography and steganalysis have complementary progress. Steganography is widely used in secret information transmission (Shi et al., 2017), watermark (Yu, 2016), copyright certification (Mun et al., 2017), forgery detection (Wolfgang & Delp, 1996) applications.

In this paper, we propose an integrated steganography and steganalysis framework with generative adversarial networks, and use ISS-GAN to represent the method in this paper. (ISS is the acronym of integrated steganography and steganalysis.) ISS-GAN combines the steganalysis's evaluation metrics of secure steganography with the advantages in latest GAN principle, and integrate the counterparts into single framework. Firstly, we will simulate the steganalysis process with discriminative model. It will help us to dynamically change the capacity of cover images, and understand their sensitivity to semantic change. Then with the fine-tuning adversarial training process of steganography generative model and steganalysis discriminative model, ISS-GAN can iteratively reduce the consistent loss between original cover images and generated stego images. Finally, when ISS-GAN gets the minimal consistent differences, the generated stego images can hardly be distinguished from original cover images. In the training process, we also involve some intentional attacks (noise, compression, etc.) in dataset. The mixture of training dataset can further improve the security of ISS-GAN. By comparing ISS-GAN with the state-of-the-art steganography methods in benchmark datasets, we can conclude that ISS-GAN has the advantages in improving the quality and security of generated stego images.

In Figure 1, can you differentiate between Van Gogh's paintings in (a) and (b)? Or Monet's paintings in (e) and (f)? Actually, the images in (a) and (e) are the original version of drawing masters' works. The images in (b) and (f) are the stego version with ISS-GAN framework. The embedded

secret images are emblems of painters' nations: Netherland and France. The embedded info is kept imperceptible to ensure there is no influence on audience to appreciate paintings from fidelity aspect.

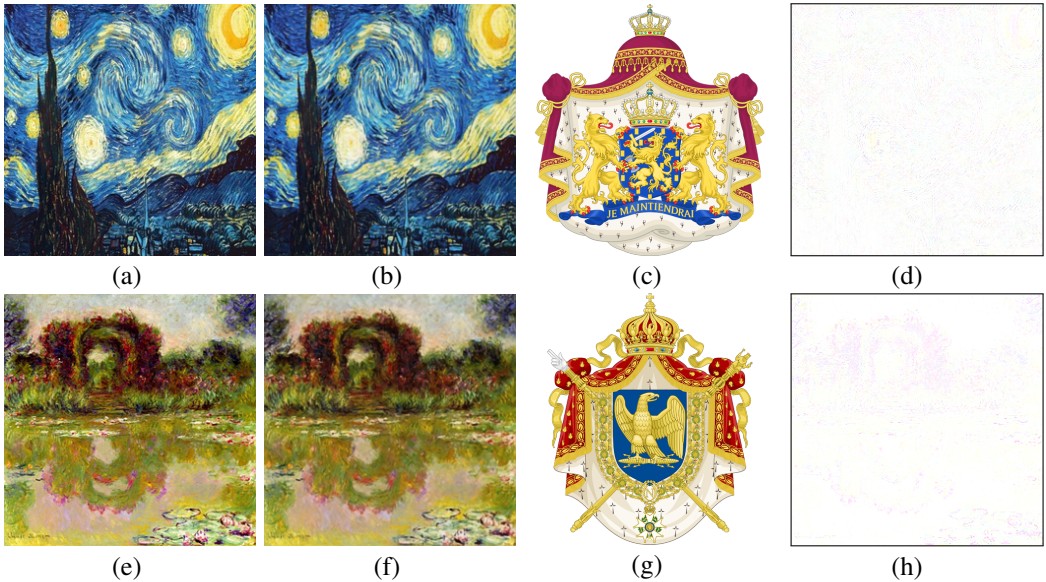

|       |       |       |       |
| :---: | :---: | :---: | :---: |
| (a)   | (b)   | (c)   | (d)   |
| (e)   | (f)   | (g)   | (h)   |

Figure 1: Illustration of ISS-GAN framework's steganographic experimental performance on the world-renowned art paintings. (a) Original version of *The Starry Night* painted by *Van Gogh*. (b) Stego version of *The Starry Night*. (c) *Emblem of Netherland* as the embedded secret image. (e) Original version of *The rose arches* painted by *Monet*. (f) Stego version of *The rose arches*. (g) *Emblem of France* as the embedded secret image. (d,h) Residual difference between original and stego versions. (We inverse the color to emphasize the difference.)

## 2 RELATED WORK

State-of-the-art steganography approaches can be categorized into three types.

**Least Significant Bit Steganography**  The main strength of this category is that algorithms are theoretically simple and have low computational complexities. Secret information is embedded into cover image with the operations like shifting or replacing of pixels. In typical Least Significant Bit (LSB) algorithm, pixel values of cover image and secret messages are represented by binary form. Stego image generation process is implemented by replacing the least significant bits of cover image with the most significant bits of secret information. In (Das et al., 2018), authors proposed to generate a LSB based hash function for image authentication process, which can provide good imperceptibility between original image and stego image with hash bits. Moreover, it can successfully identify tamper by a process of tamper localization.

**Content Adaptive Steganography**  In this category, some sophisticated steganographic algorithms design a hand-crafted distortion function which is used for selecting the embedding localization of the image. These algorithms are the most secure image steganography in spatial domain, such as Wavelet Obtained Weights (WOW), Highly Undetectable Steganography (HUGO), S-UNIWARD, etc. WOW (Holub & Fridrich, 2012) embeds information into the cover image according to textural complexity of regions. In WOW algorithm, the more texturally complex the image region is, the more pixel values will be modified in this region. HUGO (Pevnỳ et al., 2010) defines a distortion function domain by assigning costs to pixels based on the effect of embedding some information within a pixel. It uses a weighted norm function to represent the feature space. S-UNIWARD (Holub et al., 2014) proposes a universal distortion function that is independent of the embedded domain. Despite the diverse implementation details, the ultimate goals are identical in this category. They are all devoted to minimize distortion functions, to embed the secret into the noisy area or complex textures, and to avoid the smooth regions of the cover images.

**Deep Learning based Steganography**  As deep learning has brilliant capability in image processing and generation, researchers also attempt to utilize it in steganography. (Volkhonskiy et al., 2017)

introduces a new model for generating more steganalysis-secure cover images based on deep convolutional generative adversarial networks. (Dong et al., 2018) proposes a steganography model which can conceal a gray secret image into a color cover image with the same size, and generate stego image which seems quite similar to cover image in semantics and color. (Shi et al., 2017) wants to generate more secure covers for steganography. Based on Wasserstein GAN (Arjovsky et al., 2017), the proposed algorithm is efficient to generate cover images with higher visual quality.

## 3 FRAMEWORK OF ISS-GAN

### 3.1 PRINCIPLE OF ISS-GAN

In the proposal, ISS-GAN is a steganography framework to embed secret message into the source cover image. So here are two essential metrics to evaluate the steganographic algorithm.

- Secret info should remain imperceptible until it is extracted by specific authorized receiver.
- Stego image should be secure and intact to resist tampering and attacks.

In traditional state-of-the-art frameworks, the imperceptibility is achieved by carefully choosing the LSB in pixel domain, or relied on hand-crafted distortion function in traditional steganography. So the features and algorithms need meticulous artificial design. Moreover, these designs heavily rely on the characteristics of target images. So it is very hard for these schemes to become general solutions in various applications. The artificial designed features are also vulnerable to intentional and hybrid attacks. For deep learning based steganography, the main focus is to generate the steganalysis-secure cover images. But in many real applications, the cover images are given. So how to fully utilize the given images to hide secret, and to improve the security of generated stego images are not answered.

After analysing the drawbacks of state-of-the-art algorithms, we find GAN is very suitable for integrated steganography and steganalysis framework. Instead of artificial design, the generative network can learn from training samples and generate the suitable imperceptible features by itself. The discriminative network can simulate the function of steganalysis. The iterative adversarial training process can strength the capability if steganalysis model as well as steganography model. The stronger steganalysis model will stimulate the boost of steganography model, and vice versa. Moreover, how to resist the tampering can also be learned from attacked training samples.

For the first evaluation metric, let's imagine the following situation. An eavesdropper wants to check whether the image he obtained from public media contains secret info. So he needs to discriminate the original cover image and received stego image. If these two images are perceptibly same, then the eavesdropper can hardly differentiate the stego image from the cover image.

For the purpose of steganography, we can accumulate the visual and statistic differences between cover and stego images. If the difference for each evaluation metric is small enough, we can regard this stego image as a high-quality steganography result. This aligns with the imperceptible evaluation criterion of steganography.

For the second evaluation metric, let's imagine the following situation. The eavesdropper wants to destroy the secret communication. So he makes intentional changes to the stego image, like rotate, clip, add noises and JPEG compression. Because he assume that even the image he obtained contains secret, these intentional changes will make the secret extraction method disabled. If the steganography framework is secure, and steganalysis algorithm is robust enough, the intentional changes are in vain. This aligns with the secure evaluation criterion of steganography.

GAN (Goodfellow et al., 2014) consists of the generative model and the discriminative model. The purpose of the generative model is to generate new samples which are very similar to the real samples, and attempts to confuse the discriminator. While the purpose of the discriminative model is to classify samples synthesized by the generative model and the real ones. The discriminative model will also estimate the probability that a specific sample comes from the generative model rather than the real ones. When the whole GAN model achieves Nash Equilibrium, that is to say, the generative model can generate the samples which exactly align with the character and distribution of real samples. And at the same time, the discriminative model returns the classification probability 0.5 for each pair of generated and real samples. Then this GAN model is well-trained and converged.

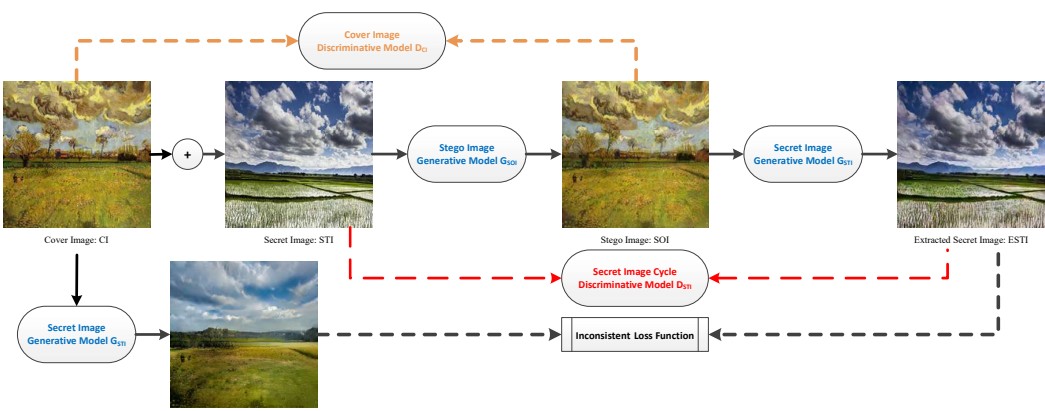

Figure 2: Framework and workflow chart of ISS-GAN

To combine the purpose of steganography, steganalysis and GAN model, we propose novel ISS-GAN framework. ISS-GAN also consists of the steganography generative and steganalysis discriminative model. The purpose of steganography generative model is to generate stego image which is aligned with the original cover image, and attempts to confuse steganalysis discriminative model. While the purpose of the discriminative model is to distinguish generated stego image from the cover image. When ISS-GAN achieves Nash Equilibrium, i.e., the generative model can generate stego image which exactly aligns with the character and distribution of cover image. And at the same time, the discriminative model returns the classification probability 0.5 for each pair of stego and cover image. This also aligns with the evaluation criterion of steganography and steganalysis. In conclusion, designing steganography and steganalysis framework is equal to make the ISS-GAN model well-trained and converged. The overall framework of ISS-GAN is shown in Figure. 2.

In ISS-GAN, there are two generative models and two discriminative models. Because steganography and steganalysis framework should contain secret info embedding and extraction processes, so it needs to learn the bijective mapping relationship between two image collections. For ISS-GAN, one image collection contains the original cover images, the other collection contains the secret images for embedding.

In the left part of Figure 2, the original cover image (*CI*) and the original secret image (*STI*) go through the stego image generative model $\boldsymbol{G}_{SOI}$, to produce the stego image (*SOI*). This is the secret embedding and stego image generation process, which can be expressed as follows.

$$SOI = \boldsymbol{G}_{SOI}(CI, STI) \tag{1}$$

In the right part of Figure 2, the stego image (*SOI*) go through the secret image generative model $\boldsymbol{G}_{STI}$, to get the extracted secret image (*ESTI*). This is the secret image extraction process, which can be expressed as follows.

$$ESTI = \boldsymbol{G}_{STI}(SOI) \tag{2}$$

The cover image discriminative model $\boldsymbol{D}_{CI}$ ensures that the distribution of images from *CI* is indistinguishable from the distribution *SOI* using an adversarial loss. This is the guarantee of the imperceptible evaluation criterion in steganography.

For the purpose of refining secret extraction, we introduce the secret image cycle discriminative model $\boldsymbol{D}_{STI}$. Because generative model is learned to transform from a source image domain to a target image domain. Take the secret image generative model $\boldsymbol{G}_{STI}$ as an example, the learned mapping relation is highly under-constrained, and cannot ensure the generated *ESTI* is indistinguishable from original *STI* (Zhu et al., 2017). So we couple this mapping relation with its inverse mapping $\boldsymbol{G}_{SOI}$, and introduce a cycle adversarial loss:

$$\boldsymbol{D}_{STI}(STI, ESTI) \to 0 \tag{3}$$

That is equal to

$$\boldsymbol{G}_{STI}(SOI) = \boldsymbol{G}_{STI}(\boldsymbol{G}_{SOI}(CI, STI)) \approx STI \tag{4}$$

Its goal is to ensure that the distribution of images from *ESTI* is indistinguishable from the distribution *STI* using cycle adversarial loss $\boldsymbol{D}_{STI}$. This is the guarantee of the secure and robust extraction criterion in steganography and steganalysis.

To refine steganalysis scheme, we introduce the extra inconsistent loss. To make the whole ISS-GAN framework useable, we should ensure the secret can only be extracted from *SOI*. If we apply the secret extraction process to *CI*, secret image should not be recovered. The inconsistent loss can be expressed as follows:

$$\max_{\boldsymbol{G}_{STI}} |\boldsymbol{G}_{STI}(CI) - \boldsymbol{G}_{STI}(SOI)| \tag{5}$$

## 3.2 Loss Function Definition

The overall loss function of ISS-GAN consists of three parts: the adversarial loss $L_{GAN}(\boldsymbol{G}_{SOI}, \boldsymbol{D}_{CI})$, the cycle adversarial loss $L_{GAN}(\boldsymbol{G}_{STI}, \boldsymbol{D}_{STI})$ and the inconsistent loss $L_{IC}$. So the loss function is written as follows:

$$L_{Overall} = L_{GAN}(\boldsymbol{G}_{SOI}, \boldsymbol{D}_{CI}) + L_{GAN}(\boldsymbol{G}_{STI}, \boldsymbol{D}_{STI}) + \lambda L_{IC}[\boldsymbol{G}_{STI}(CI), \boldsymbol{G}_{STI}(SOI)], \tag{6}$$

where $\lambda$ is the parameter to adjust the percentages between adversarial loss and inconsistent loss. The inconsistent loss needs to change to the minimization format as follows.

$$\min_{\boldsymbol{G}_{STI}} \frac{1}{|\boldsymbol{G}_{STI}(CI) - \boldsymbol{G}_{STI}(SOI)|} \tag{7}$$

In ISS-GAN framework, the quality of generated stego image *SOI* and extracted secret image *ESTI* is judged by the difference from original cover image *CI* and original secret image *STI*, respectively. In this paper, two quantitative image effect indicators are applied to measure the differences (Yu, 2017). Peak Signal to Noise Ratio (PSNR) indicator is applied to assess the effect difference from the gray-level fidelity aspect. Structural Similarity (SSIM) (Wang et al., 2004) indicator which is an image quality assessment indicator based on the human vision system is applied to assess the effect difference from the structure-level fidelity aspect. The definitions of these two evaluation indicators are as follows.

$$PSNR(\boldsymbol{x}, \boldsymbol{y}) = 10 \log_{10} \left( \frac{(MAX_I)^2}{MSE(\boldsymbol{x}, \boldsymbol{y})} \right), \tag{8}$$

where $MAX_I$ is the maximum possible pixel value of images: $\boldsymbol{x}$ and $\boldsymbol{y}$. $MSE(\boldsymbol{x}, \boldsymbol{y})$ represents the Mean Squared Error (MSE) between images: $\boldsymbol{x}$ and $\boldsymbol{y}$.

$$SSIM(\boldsymbol{x}, \boldsymbol{y}) = \frac{(2\mu_{\boldsymbol{x}}\mu_{\boldsymbol{y}} + C_1)(2\sigma_{\boldsymbol{xy}} + C_2)}{(\mu_{\boldsymbol{x}}^2 + \mu_{\boldsymbol{y}}^2 + C_1)(\sigma_{\boldsymbol{x}}^2 + \sigma_{\boldsymbol{y}}^2 + C_2)}, \tag{9}$$

where $\mu_{\boldsymbol{x}}$ and $\mu_{\boldsymbol{y}}$ represent the average grey values of images. Symbol $\sigma_{\boldsymbol{x}}$ and $\sigma_{\boldsymbol{y}}$ represent the variances of images. Symbol $\sigma_{\boldsymbol{xy}}$ represents covariance between images. $C_1$ and $C_1$ are two constants which are used to prevent unstable results when either $\mu_{\boldsymbol{x}}^2 + \mu_{\boldsymbol{y}}^2$ or $\sigma_{\boldsymbol{x}}^2 + \sigma_{\boldsymbol{y}}^2$ is very close to 0.

## 3.3 ISS-GAN Network Structure

For ISS-GAN, the resolution of cover image *CI* and secret image *STI* is 256×256. The network structure of stego image generative model $\boldsymbol{G}_{SOI}$ includes a convolution layer (kernel size = 7, stride = 0, pad = 0), two convolution layers (kernel size = 3, stride = 2, pad = 1), nine residual blocks (He et al., 2016), and two deconvolution layers (kernel size = 3, stride = 2, pad = 1, outside pad = 1), and a convolution layer (kernel size = 7, stride = 0, pad = 0). Each convolution and deconvolution layer follows with an instance normalization layer and a ReLU layer. The structure of secret image generative model $\boldsymbol{G}_{STI}$ is identical with $\boldsymbol{G}_{SOI}$.

The network structure of cover image discriminative model $\boldsymbol{D}_{CI}$ is similar with PatchGAN model (Isola et al., 2017). Each time, it operates a image patch with 70×70 size, and classifies whether this patch is real or fake. The model will run across the whole image, and average all results in the 70×70 overlapping patches to provide the ensemble output. The architecture of such a patch-level discriminative model requires fewer parameters and runs faster than a full-image discriminator (Yi et al., 2017). Moreover, it has no constraints over the size of the input image. $\boldsymbol{D}_C$ contains a convolution layer (kernel size = 4, stride = 2, pad = 1) follows with a leaky ReLU layer, three convolution layers (kernel size = 4, stride = 2, pad = 1) follows with an instance normalization layer and a leaky ReLU layer, a convolution layer (kernel size = 4, stride = 1, pad = 1) follows with an instance normalization layer and a leaky ReLU layer, a convolution layer (kernel size = 4, stride = 1, pad = 1)

follows with a sigmoid layer to output a scalar output between [0, 1]. The structure of secret image cycle discriminative model $\boldsymbol{D}_{STI}$ is identical with $\boldsymbol{D}_{CI}$.

Moreover, to improve the convergence performance, we use Adam optimizer (Kinga & Adam, 2015) instead of stochastic gradient descent (SGD) optimizer. In practice, Adam optimizer can be adaptive to the training of ISS-GAN. It is computationally efficient and has little memory requirements. The hyper-parameters of Adam optimizer are: $\beta_1$=0.5, $\beta_2$=0.999. The base learning rate is 0.0002.

## 4 EXPERIMENTAL RESULTS

### 4.1 STEGANOGRAPHY PERFORMANCE EXPERIMENTS

In the secret embedding and stego image generation performance experiments, we adopt the benchmark images as the cover images *CI* shown in the first row of Figure 3 to test the performance of proposed ISS-GAN framework. The embedded secret image used is the Barbara benchmark image. We use PyTorch as the framework and train ISS-GAN with 150 epochs.

The generated stego images *SOI* are shown in the second row of Figure 3. For illustration purpose, the residual differences between cover and stego images are shown in the third row of Figure 3. The *PSNR* and *SSIM* metrics for generated stego images *SOI* versus cover images *CI* are shown in Table 1. (*SOI* is used as image $\boldsymbol{x}$, and *CI* is used as image $\boldsymbol{y}$ for *PSNR* and *SSIM* metrics calculation equations (8) and (9).) The results shown in Figure 3 and Table. 1 can prove the high quality and difference imperceptibility of *SOI* in qualitative and quantitative aspects.

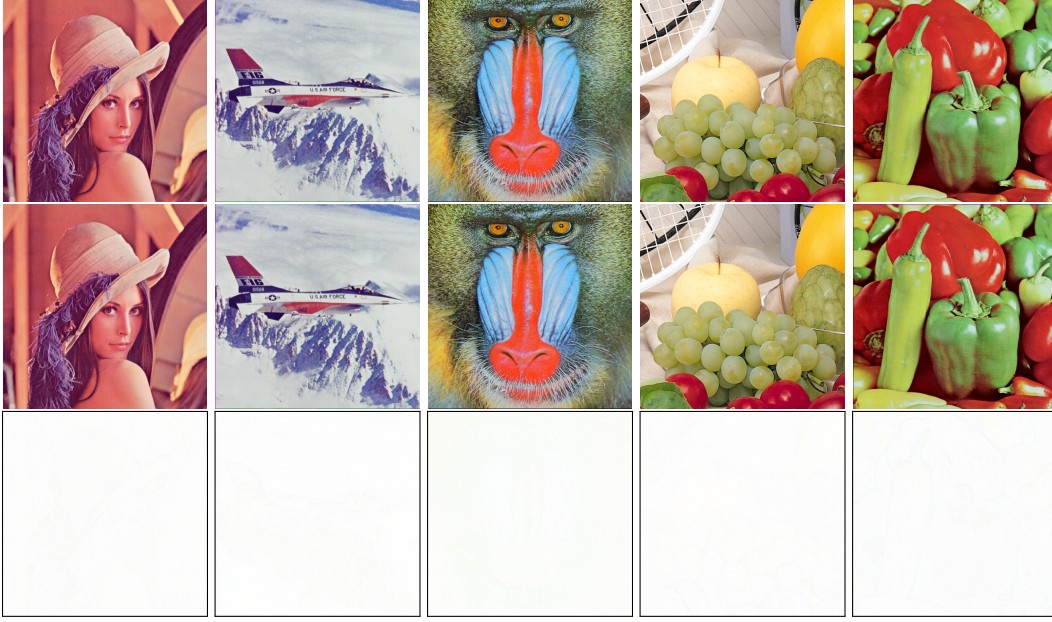

Figure 3: Stego images *SOI* generation performance of ISS-GAN. Row 1: Original cover images: Lena, Airplane, Baboon, Fruits and Peppers, Row 2: Corresponding generated stego images, Row 3: Residual difference between original and stego images. (We inverse the color to emphasize the difference. Because the differences are inconspicuous. Please magnify to see the differences which mainly on the marginal parts of objects.)

Table 1: Evaluation metrics of generated stego images *SOI*

| Metrics/Images | Lena | Airplane | Baboon | Fruits | Peppers |
|---|---|---|---|---|---|
| PSNR | 33.0170 | 33.0065 | 29.1163 | 33.9085 | 30.5124 |
| SSIM | 0.9390 | 0.9589 | 0.9335 | 0.9510 | 0.9034 |

Let's have a further analysis of the obtained results. If we magnify Figure 3 to see the residual differences, we can find they are mainly on the marginal and textural parts of objects. For example, the hat of Lena, the edges of F16 plane, the skin and whiskers of baboon, the profile of fruits and peppers, etc. It means ISS-GAN tends to hide the secret info into marginal parts of the object in

original cover images. In information theory, textures and edges represent the high frequency parts of the image, while smooth regions represent the low frequency parts of the image. If we change the low frequency parts, it is easy to be detected by steganalysis method. So many state-of-the-art steganography algorithms transform the cover image from spatial domain to frequency domain. Change the tiny part in high frequency parts, and transform it back to spatial domain. Moreover, when we discuss the state-of-the-art content adaptive steganography algorithms, we find the ultimate goal is trying to embed the secret image into the parts with complex edges and textures, and avoiding the smooth regions of the cover images. The behavior of ISS-GAN is very similar to the state-of-the-art steganography algorithms. But the state-of-the-art algorithms need to design a hand-crafted distortion function to achieve the goal, while ISS-GAN learns from the discriminative network which simulates the behaviors of steganalysis. From the learning process, the generative network in ISS-GAN finds steganalysis method are very sensitive to the low frequency parts, and not so sensitive to the high frequency parts. So the stego images generated by ISS-GAN generative network mainly hide their secret info into marginal and textural parts to ensure the best imperceptibility.

### 4.2 STEGANALYSIS QUALITATIVE PERFORMANCE EXPERIMENTS

In the steganalysis qualitative experiments, we adopt the world-renowned art paintings shown in Figure 1 to test the performance of proposed ISS-GAN and its robustness to different patterns of noise attack. Several patterns of noises are adopted respectively to imitate real-world noise attacks. We use PyTorch as the framework and train ISS-GAN with 200 epochs. The extracted secret image *ESTI* are shown in Figure 4 and Appendix Figure 8. The results shown in Figure 4 and 8 can prove the high quality of *ESTI* in qualitative aspect.

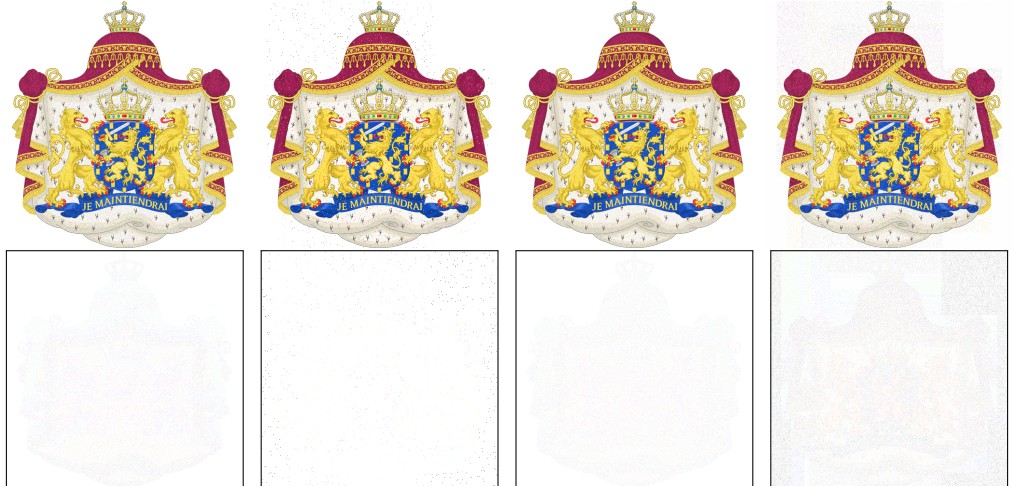

Figure 4: Extracted secret images if stego images are attacked by noise. Column 1: Multiplicative noise, Column 2: Salt and pepper noise, Column 3: Gaussian white noise, Column 4: Poisson noise. Row 2 and 4: Residual difference between embedded secret images and extracted secret images. (We inverse the color to emphasize the difference.)

### 4.3 STEGANALYSIS QUANTITATIVE COMPARATIVE EXPERIMENTS

We compare ISS-GAN with the state-of-the-art steganography methods in various image benchmarks. Here we use steganalysis process to extract secret images from stego images, and evaluate the security criteria of steganography algorithms. For LSB steganography algorithms, we choose *LSB-TLH* Das et al. (2018). For content adaptive steganography algorithms, we choose *WOW* Holub & Fridrich (2012), *HUGO* (Pevný et al., 2010) and *S-UNIWARD* (Holub et al., 2014). For deep learning based steganography algorithms, we choose *ISGAN* Dong et al. (2018) and *SSGAN* Shi et al. (2017) for comparation. We make two group experiments. In the first group experiment, we use *Lena* as the original cover image, and the trolleybus image as the secret image. The results are shown in Figure 5. In the second group experiment, we use *Lena* as the original cover image, and the headline of ICLR conference as the secret image. The results are shown in Figure 6. In quantitative experiments, we add the JPEG compression to simulate the coding and decoding processes in real

secure information transmission system. We need to ensure ISS-GAN can work well against coding and decoding algorithms.

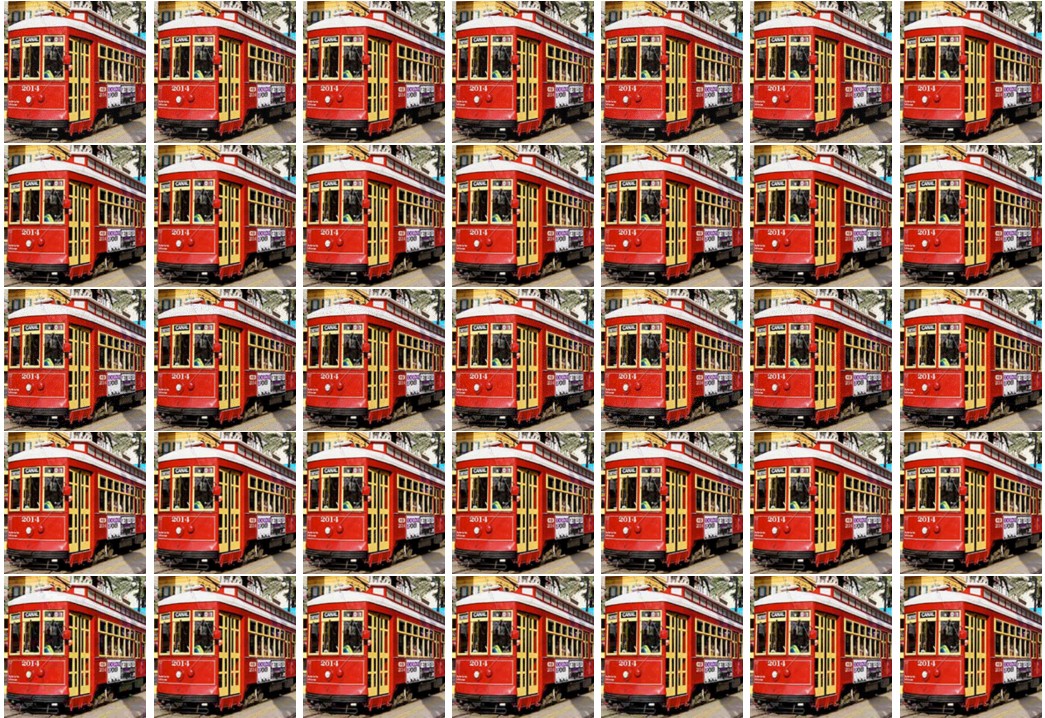

Figure 5: Extracted secret image of trolleybus after adding certain attacks. Row 1: Gaussian white noise, Row 2: Poisson noise, Row 3: Salt and pepper noise, Row 4: Speckle noise, Row 5: JPEG compression. (Results of separate Salt noise and Pepper noise are in Appendix.) Column 1-7 are *ESTI* obtained by *LSB-TLH*, *WOW*, *HUGO*, *S-UNIWARD*, *ISGAN*, *SSGAN* and proposed *ISS-GAN* algorithms.

Table 2: *PSNR* metric for extracted secret images of trolleybus

| Images/Algorithms | LSB-TLH | WOW | HUGO | S-UNIWARD | ISGAN | SSGAN | ISS-GAN |
|---|---|---|---|---|---|---|---|
| Gaussian noise | 21.9184 | 22.5196 | 25.4038 | 23.6964 | 21.0874 | 22.0219 | **28.5659** |
| Possion noise | 28.0956 | 28.1163 | 28.1097 | 28.1035 | 28.0996 | **28.1182** | 28.1181 |
| Salt & Pepper noise | 20.6125 | 22.3013 | 22.3739 | 22.1683 | 20.2038 | 21.0407 | **23.3684** |
| Salt noise | 19.7074 | 20.2493 | 22.0935 | 20.4663 | 19.6455 | 19.7446 | **22.8699** |
| Pepper noise | 21.8021 | 22.4331 | 22.5555 | 24.2972 | 21.7754 | 21.8409 | **25.0980** |
| Speckle noise | 27.9778 | 33.8242 | 32.7279 | 30.5234 | 28.8135 | 29.6587 | **37.5805** |
| JPEG Compression | 26.6862 | 30.4574 | 31.0763 | 29.7196 | 27.9319 | 28.8819 | **31.6285** |

Table 3: *SSIM* metric for extracted secret images of trolleybus

| Images/Algorithms | LSB-TLH | WOW | HUGO | S-UNIWARD | ISGAN | SSGAN | ISS-GAN |
|---|---|---|---|---|---|---|---|
| Gaussian noise | 0.6548 | 0.6802 | 0.7885 | 0.7271 | 0.6182 | 0.6589 | **0.8772** |
| Possion noise | 0.8876 | 0.8878 | 0.8878 | 0.8877 | 0.8874 | **0.8880** | **0.8880** |
| Salt & Pepper noise | 0.7338 | 0.8224 | 0.8220 | 0.8110 | 0.7190 | 0.7584 | **0.8464** |
| Salt noise | 0.7147 | 0.7384 | 0.8096 | 0.7476 | 0.7127 | 0.7156 | **0.8352** |
| Pepper noise | 0.8119 | 0.8295 | 0.8333 | 0.8780 | 0.8100 | 0.8124 | **0.8941** |
| Speckle noise | 0.8955 | 0.9635 | 0.9542 | 0.9304 | 0.9062 | 0.9206 | **0.9836** |
| JPEG Compression | 0.8909 | 0.9425 | 0.9487 | 0.9343 | 0.9304 | 0.9245 | **0.9538** |

The *PSNR* and *SSIM* metrics for extracted secret image of trolleybus and ICLR conference headline are shown in Table 2~ 5, respectively. In Table 2~ 5, extracted secret image is used as image $x$, and original secret image is used as image $y$ for *PSNR* and *SSIM* metrics calculation equations (8) and (9). According to these metrics, the security of ISS-GAN outperforms all other state-of-the-art steganography algorithms in quantitative aspect.

In these two group experiments, *PSNR* and *SSIM* metrics, the closest competitors are content adaptive steganography algorithms. *WOW*, *HUGO* and *S-UNIWARD* have quite good performance on steganography security. But for *PSNR* metric, *ISS-GAN* can still achieve 1.01X~1.12X relative improvement over the second highest *PSNR* algorithms with trolleybus secret image embedded, and

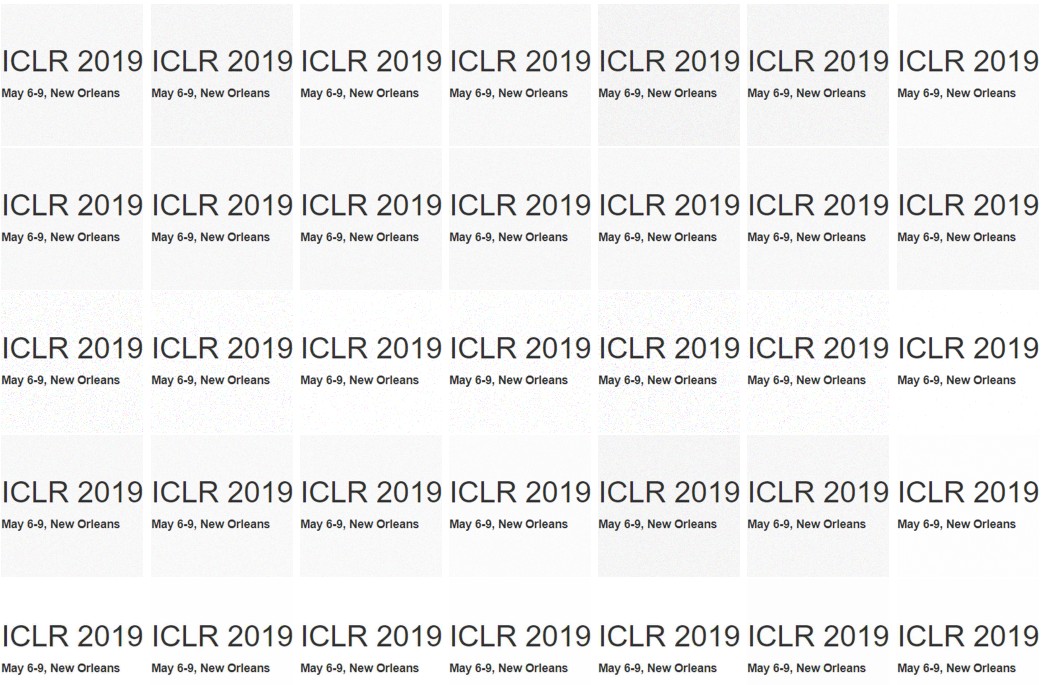

Figure 6: Extracted secret image of ICLR conference headline after adding certain attacks. Row 1: Gaussian white noise, Row 2: Poisson noise, Row 3: Salt and pepper noise, Row 4: Speckle noise, Row 5: JPEG compression. (Results of separate Salt noise and Pepper noise are in Appendix.) Column 1-7 are *ESTI* obtained by *LSB-TLH*, *WOW*, *HUGO*, *S-UNIWARD*, *ISGAN*, *SSGAN* and proposed *ISS-GAN* algorithms.

Table 4: *PSNR* metric for extracted secret images of ICLR conference headline

| Images/Algorithms | LSB-TLH | WOW | HUGO | S-UNIWARD | ISGAN | SSGAN | ISS-GAN |
|---|---|---|---|---|---|---|---|
| Gaussian noise | 25.6366 | 24.5133 | 27.9945 | 25.8188 | 22.8196 | 23.6139 | **30.0088** |
| Possion noise | 27.2302 | 27.2400 | 27.2332 | 27.2281 | 27.2352 | 27.2242 | **27.2497** |
| Salt & Pepper noise | 19.4886 | 20.8146 | 24.3997 | 21.5674 | 19.5006 | 20.7629 | **36.4338** |
| Salt noise | 30.3170 | 31.8986 | 34.0030 | 32.3590 | 31.3167 | 31.7378 | **49.0843** |
| Pepper noise | 17.3986 | 17.8621 | 18.3596 | 19.8744 | 16.4737 | 17.7617 | **35.9698** |
| Speckle noise | 26.8375 | 27.6863 | 27.8516 | 34.0109 | 24.5629 | 25.3830 | **42.8481** |
| JPEG Compression | 30.1590 | 32.1941 | 32.6464 | 31.8712 | 30.9326 | 31.1552 | **32.7724** |

achieve 1.01X∼1.81X relative improvement over the second highest *PSNR* algorithms with ICLR conference headline secret image embedded. For *SSIM* metric, *ISS-GAN* can achieve 1.01X∼1.11X relative improvement over the second highest *SSIM* algorithms with trolleybus secret image embedded, and achieve 1.01X∼1.46X relative improvement over the second highest *SSIM* algorithms with ICLR conference headline secret image embedded.

To have a further analysis of the obtained results, we can find the performance of state-of-the-art deep learning steganography algorithms is not as good as expected. The main reason is the authors are more focus on generating the new cover images which are steganalysis-secure. But in our experiments, the cover images are fixed. So we can see the performance of state-of-the-art deep learning steganography algorithms is just at the same level of LSB steganography algorithms, and is worse than content adaptive steganography algorithms.

Compare the results of first and second group experiments, we find ISS-GAN has better performance with the ICLR conference headline secret image. The amount of meaningful pixels and semantic info in ICLR conference headline image is much less than trolleybus image. This aligns with the principle of steganography, i.e., if more pixels are concealed into the cover image, then the security of stego image will be worse.

To further illustrate the effect of embedding secret info amount on the security and imperceptibility of stego image, we make the curve plots to show the quantitative experiment results of generated

Table 5: *SSIM* metric for extracted secret images of ICLR conference headline

| Images/Algorithms | LSB-TLH | WOW | HUGO | S-UNIWARD | ISGAN | SSGAN | ISS-GAN |
|---|---|---|---|---|---|---|---|
| Gaussian noise | 0.7163 | 0.6647 | 0.8084 | 0.7228 | 0.5808 | 0.6200 | **0.8682** |
| Possion noise | 0.7706 | 0.7707 | 0.7710 | 0.7709 | 0.7709 | 0.7703 | **0.7716** |
| Salt & Pepper noise | 0.6566 | 0.7352 | 0.8839 | 0.7664 | 0.6569 | 0.7362 | **0.9912** |
| Salt noise | 0.9900 | 0.9927 | 0.9951 | 0.9931 | 0.9918 | 0.9924 | **0.9998** |
| Pepper noise | 0.5160 | 0.5455 | 0.5834 | 0.6775 | 0.4464 | 0.5428 | **0.9900** |
| Speckle noise | 0.7422 | 0.7718 | 0.7790 | 0.9340 | 0.6461 | 0.6752 | **0.9904** |
| JPEG Compression | 0.9841 | 0.9888 | 0.9892 | 0.9880 | 0.9863 | 0.9877 | **0.9897** |

stego images *SOI* versus cover images *CI* as shown in Figure 7. Here, we use the pixel-ratio to control the amount of embedding secret info. It is defined as the ratio of valid pixels amount in secret image versus those in cover images. For example, if the amount of valid pixels in $256 \times 256$ size cover image is 63000, and the amount of valid pixels in secret image is 15000, then pixel-ratio is 0.2381.

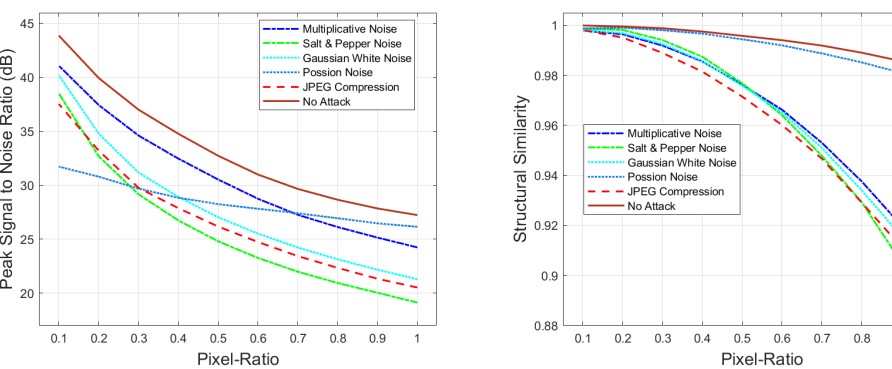

Figure 7: *PSNR* and *SSIM* metrics for generated stego images *SOI* versus cover images *CI* with different pixel-ratio and attacks.

From the curve plots shown above, we can see *PSNR* and *SSIM* metrics decline with the increase of pixel-ratio. Under the noise attack or image compression, *PSNR* and *SSIM* metrics are worse than the situations without attack, and also decline with the increase of pixel-ratio. The detailed results further prove the inherent contradiction between embedded secret amount and security of stego image. So in the real applications, ISS-GAN should make the trade-off between embedding capacity, imperceptibility and security according to real requirements, just like all state-of-the-art steganography methods. This curve can tell user the largest embedded secret capacity at certain imperceptibility and security level. So it is helpful for user to choose the most suitable embedded secret image in real secure information transmission systems. For example, if the user want to generate a stego image with no less than 25dB *PSNR* and 0.97 *SSIM* versus cover image. Considering the noise attacks and image compression possibility, the largest embedded secret pixel-ratio should be less than 0.5.

## 5 CONCLUSION AND FUTURE WORKS

In this paper, we integrate steganography and steganalysis into single framework. The good performance of ISS-GAN derives from the following factors.

- The discriminative network simulates the features of steganalysis. It helps to understand the sensitivity of cover images to semantic changes.
- The introduction of cycle discriminative model and inconsistent loss helps to enhance the quality and security of generated stego image.
- The mixture training dataset can further improve the robustness and security of ISS-GAN framework. How to resist the tampering can be learned from attacked training samples.
- The iterative adversarial training process can strength the capability if steganalysis model and steganography model at the same time. The stronger steganalysis model will stimulate the improvement of steganography model, and vice versa.

In the future, we will study the influence of color cover/secret image and gray cover/secret image to the proposed ISS-GAN framework.

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

# 6 APPENDIX

## 6.1 INTRODUCTION OF STEGANOGRAPHY AND STEGANALYSIS

Imagine that you are an enthusiastic shutterbug. You are on trip to New Orleans and took a nice photo of St. Louis Cathedral. You want to send the beautiful photo and reveal the romantic feelings to your girlfriend. As your girlfriend is the Ph.D candidate of computer vision, so you want to share the romantic words in her professional manner. You embed the words by changing the least significant bits of the photograph, because this method can hide the romantic words in nearly invisible way. As a social media network fan, you share this photo on Facebook. Many friends leave the messages to express their love of this photo. And your girlfriend write the following sentence under your photo. "Wonderful photo. By the way, I like the clever idea. I think I am the first audience who have read the hidden words. I love you, too." You will be in a cheerful mood, and appreciate the clever communication method that only you and your girlfriend can "see" the secret information inside the photograph.

This is a simple scenario to show the basic workflow of steganography and steganalysis. Steganography is defined as the art and science of hiding information in ways that prevent the detection of hidden messages (Obaid, 2015). Steganography literally means "covered writing" and is usually interpreted to hide information in other information. In the simple scenario aforementioned, you apply steganography to hide the romantic information into the photo of New Orleans. The romantic information is called the secret message, while the original photo of New Orleans is called the cover image. As the counterpart, the main idea of steganalysis is to analyze whether the received information contains any hidden information, and to recover the hidden information if possible (Volkhonskiy et al., 2017). In the simple scenario aforementioned, the social network plays the role of the public channel (Hayes & Danezis, 2017), and the posted photo is called stego image (Dong et al., 2018), which contains the secret message. Your girlfriend applies steganalysis to discover and recover the secret information you embedded. Since their birth, steganography and steganalysis have complementary progress.

## 6.2 EXPERIMENT RESULTS

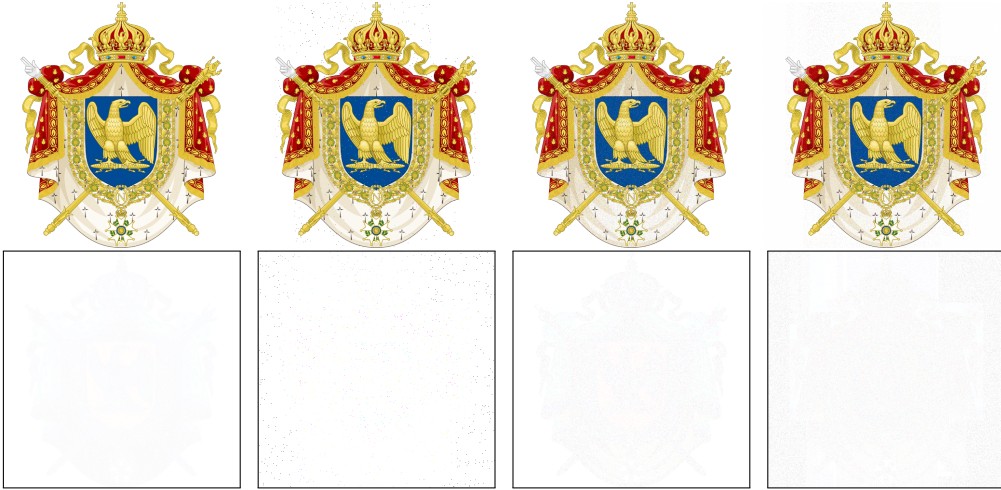

Figure 8: Extracted secret images if stego images are attacked by noise. Column 1: Multiplicative noise, Column 2: Salt and pepper noise, Column 3: Gaussian white noise, Column 4: Poisson noise. Row 2 and 4: Residual difference between embedded secret images and extracted secret images. (We inverse the color to emphasize the difference.)

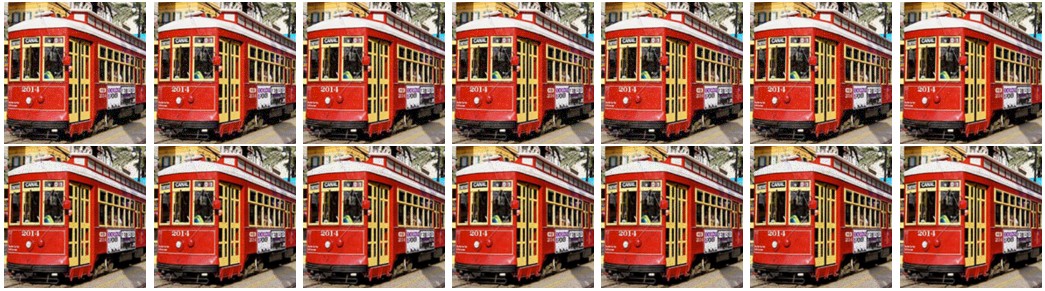

Figure 9: Extracted secret image of trolleybus after adding certain attacks. Row 1: Salt noise, Row 2: Pepper noise. Column 1-7 are *ESTI* obtained by *LSB-TLH*, *WOW*, *HUGO*, *S-UNIWARD*, *ISGAN*, *SSGAN* and proposed *ISS-GAN* algorithms.

ICLR 2019 ICLR 2019 ICLR 2019 ICLR 2019 ICLR 2019 ICLR 2019 ICLR 2019
May 6-9, New Orleans   May 6-9, New Orleans   May 6-9, New Orleans   May 6-9, New Orleans   May 6-9, New Orleans   May 6-9, New Orleans   May 6-9, New Orleans

ICLR 2019 ICLR 2019 ICLR 2019 ICLR 2019 ICLR 2019 ICLR 2019 ICLR 2019
May 6-9, New Orleans   May 6-9, New Orleans   May 6-9, New Orleans   May 6-9, New Orleans   May 6-9, New Orleans   May 6-9, New Orleans   May 6-9, New Orleans

Figure 10: Extracted secret image of ICLR conference headline after adding certain attacks. Row 1: Salt noise, Row 2: Pepper noise. Column 1-7 are *ESTI* obtained by *LSB-TLH*, *WOW*, *HUGO*, *S-UNIWARD*, *ISGAN*, *SSGAN* and proposed *ISS-GAN* algorithms.

