# OpenReview forum: "Integrated Steganography and Steganalysis with Generative Adversarial Networks"
_ICLR.cc/2019/Conference_

### Official Review · AnonReviewer2 · 2018-11-02
**eye-catching image manipulation but lack justification in real use**

**Rating:** 5
**Confidence:** 2

**Review:**

 This paper proposed a so-called ISS-GAN framework for data hiding in images, which  integrates steganography and steganalysis processes in GAN. The discriminative model simulate the steganalysis process, and the steganography generative model is to generate stego image, and confuse steganalysis discriminative model.

Overall the application seems interesting. My concern is its use in real secure information transmission systems: it can fool human eyes but what is its capacity against decoding algorithms; if the intent is to transmit some hidden information, how the receiver is supposed to decode it; is there something similar to the public key in encryption systems? These basic questions/concepts should be made clear to the reader to avoid confusion.

The evaluation protocol should be clarified and especially on how the PSNR is calculated (i.e., using the reconstructed secret image and real one?)

---

### Official Review · AnonReviewer3 · 2018-11-04
**GAN for steganography: interesting but hard to judge**

**Rating:** 6
**Confidence:** 4

**Review:**

The paper is on applying GAN to steganography, and it is especially suited for people who does not know steganography. In fact, 1/4 of the paper is spent on introducing (in a very clear and catchy way) the basics. State of the art section nonetheless shows that the authors are grounded with the very last related work. The novel GAN framework is called ISS-GAN. Simply speaking, ISS-GAN is built by the combo: <steganography generative, steganalysis discriminative>. In the details, the implementation is not straightforward, and embeds interesting ideas. Two generative and two discriminative models are intertwined: one generative for the stego image generation process, the other generative for the secret image extraction process. The two discriminative models are for ensuring that the distribution of cover images is indistinguishable from the stego distribution and to ensure that the distribution of extracted secret images is indistinguishable from the distribution of the original secret images. In rough terms, the adversarial double loop serves to ensure that the embedding and the extraction function preserve both the aspect of the original and the secret images. PSNR and SSIM metrics are employed to give a quantitative check of the approach.
Obvious questions are about the scalability of the approach, since it applies to 256x256 miniatures, which act well on the figures of an ICLR paper but poor in reality. At the same time, is it reasonable to show images such that Figure 3? It is obvious stega processes are now very effective, and stressing that by showing seemingly same images it is not very illustrative. Figure 4 typos are reated to row column ordering, but even in this case is hard to spot something without spending time in magnifying the pdf playing at look at the differences… Figure 5 and 6 are even worse, with tens of practically equal images. Less images with differences pictures should have been better. In addition, with these qualitative results, it is hard to give a weight to the quantitative comparisons, in terms of both the PSNR and SSIM metrics.
The most obvious and perhaps elementary question I have is how risky is that the ISS GAN model can be replicated by an attacker which sniffs the secret images by simply having a training set which approximatively is the same of the primary sender?
The other questions are about how much realistic is that the images are affected by the noises of Table 5, nowadays?

---

### Official Review · AnonReviewer1 · 2018-11-04
**Work uses Generative Adversarial Networks (GAN) paradigm to achieve Steganography/Steganalysis but  fails to dig deeper into obtained results to explain suitability of GAN for Steganography/Steganalysis or the types of embedded images. In need of serious rewriting.**

**Rating:** 5
**Confidence:** 5

**Review:**

Paper uses Generative Adversarial Networks' (GAN) paradigm to achieve Steganography/Steganalysis but  fails to dig deeper into obtained results to explain suitability of GAN for Steganography/Steganalysis or the types of embedded images (textures, patterns, complexity etc.) most suitable.

- Figure 3/Page 7: Providing the residual differences between covers and stego-images would be appreciated.
- Define Acronyms before using them: ISS-GAN for example (Integrated Steganography-Steganalysis (ISS)).
- General concepts like State of the Art should not in our humble opinion be used through acronyms (SOTA).
 - Page 3/Paragraph (Deep Learning based Steganography):  Clean your references:
     - Paper (Volkhonskiy et al.,2017) , Paper (Dong et al., 2018), Paper (Shi et al.,2017).

In need of a serious rewriting:
 - No need for 1. Introduction's first paragraph. Good intention for a simplistic definition scenario rendered redundant by following paragraph.
 - Multiple needless repetitions especially for definitions such as those of Steganography/Steganalysis (Paragraphs 4 & 5, page 4).
 - Ill-articulated phrases:
    - Introduction: - 'Steganalysis as the counterpart, is an attack to the steganography'.
                               - 'Since their birth, steganography and steganalysis promote the progress of each other.' Just say complementary processes.
    - Paragraph 3/Page3: 'Their model is suitable for embedding secret with the random key.'.

In need of more proofing (Phrasing errors):
  - Abstract: 'Its application on data generation' instead of 'It’s application on data generation'.
  - Introduction: 'Imagine that you are' instead of 'Imaging that you are'.
  - Multiple unjustified uses of  'the':
       - we propose 'the' novel framework.
       - You are on 'the' trip to New Orleans

---

### Author Response · Authors · 2018-12-04
**OpenReview Compare tool cannot work here**

The revision compare tool is not work with the error "Unable to compare".
Have reported to ICLR Program Chairs and OpenReview. Still wait for their fix.
But we have checked that the latest version modified: 26 Nov 2018, 19:00 is our revised version according to reviewers' comments.

---

> ### Public Comment · ~Melisa_Bok1 · 2018-12-05
> **Max size file should be 40mb**
>
> Hi Authors,
>
> I'm Melisa from the OpenReview team. Our tool to compare pdfs is using a third party api that supports files up to 40mb of size. Please try to reduce the size of the file and we can help you to replace the current versions.
>
> Best,
>
> Melisa

---

### Meta-Review · Area_Chair1 · 2018-12-13
**promising results but hard to assess due to lack of clarity**

**Confidence:** 4
**Recommendation:** Reject

**Metareview:**

1. Describe the strengths of the paper.  As pointed out by the reviewers and based on your expert opinion.

- The problem and approach, steganography via GANs, is interesting.
- The results seem promising.

2. Describe the weaknesses of the paper. As pointed out by the reviewers and based on your expert opinion. Be sure to indicate which weaknesses are seen as salient for the decision (i.e., potential critical flaws), as opposed to weaknesses that the authors can likely fix in a revision.

The original submission was imprecise and difficult to follow and, while the AC acknowledges that the authors made significant improvements, the current version still needs some work before it's clear enough to be acceptable for publication.

3. Discuss any major points of contention. As raised by the authors or reviewers in the discussion, and how these might have influenced the decision. If the authors provide a rebuttal to a potential reviewer concern, it’s a good idea to acknowledge this and note whether it influenced the final decision or not. This makes sure that author responses are addressed adequately.

Concerns varied by reviewer and there was no main point of contention.

4. If consensus was reached, say so. Otherwise, explain what the source of reviewer disagreement was and why the decision on the paper aligns with one set of reviewers or another.

The reviewers did not reach a consensus. The final decision is aligned with the less positive reviewers, one of whom was very confident in his/her review. The AC agrees that the paper should be made clearer and more precise.